# Neuroprotective Effects of a Novel Inhibitor of c-Jun N-Terminal Kinase in the Rat Model of Transient Focal Cerebral Ischemia

**DOI:** 10.3390/cells9081860

**Published:** 2020-08-08

**Authors:** Mark B. Plotnikov, Galina A. Chernysheva, Vera I. Smolyakova, Oleg I. Aliev, Eugene S. Trofimova, Eugene Y. Sherstoboev, Anton N. Osipenko, Andrei I. Khlebnikov, Yana J. Anfinogenova, Igor A. Schepetkin, Dmitriy N. Atochin

**Affiliations:** 1Department of Pharmacology, Goldberg Research Institute of Pharmacology and Regenerative Medicine, Tomsk NRMC, 3 Lenin ave, 634028 Tomsk, Russia; bona2711@mail.ru (G.A.C.); light061265@mail.ru (V.I.S.); oal67@yandex.ru (O.I.A.); eugenie76@mail.ru (E.S.T.); sherstoboev_eu@pharmso.ru (E.Y.S.); 2National Research Tomsk State University, 36 Lenin ave., 634050 Tomsk, Russia; 3Department of Pharmacology, Siberian State Medical University, 2 Moskovskiy tract, 634050 Tomsk, Russia; osipenko-an@mail.ru; 4Kizhner Research Center, Tomsk Polytechnic University, 634050 Tomsk, Russia; aikhl@chem.org.ru (A.I.K.); igor@montana.edu (I.A.S.); atochin@cvrc.mgh.harvard.edu (D.N.A.); 5Cardiology Research Institute, Tomsk NRMC, 111a Kievskaya St., 634012 Tomsk, Russia; cardio.intl@gmail.com; 6Department of Microbiology and Immunology, Montana State University, Bozeman, MT 59717, USA; 7Cardiovascular Research Center, Cardiology Division, Massachusetts General Hospital, Harvard Medical School, Charlestown, MA 02129, USA

**Keywords:** neuroprotection, inhibitor of c-Jun N-terminal kinase, 11*H*-indeno[1,2-*b*]quinoxalin-11-one oxime sodium salt, focal cerebral ischemia-reperfusion

## Abstract

A novel specific inhibitor of c-Jun N-terminal kinase, 11*H*-indeno[1,2-*b*]quinoxalin-11-one oxime sodium salt (IQ-1S), has a high affinity to JNK3 compared to JNK1/JNK2. The aim of this work was to study the mechanisms of neuroprotective activity of IQ-1S in the models of reversible focal cerebral ischemia (FCI) in Wistar rats. The animals were administered with an intraperitoneal injection of IQ-1S (5 and 25 mg/kg) or citicoline (500 mg/kg). Administration of IQ-1S exerted a pronounced dose-dependent neuroprotective effect, not inferior to the effects of citicoline. Administration of IQ-1S at doses of 5 and 25 mg/kg reduced the infarct size by 20% and 50%, respectively, 48 h after FCI, whereas administration of citicoline reduced the infarct size by 34%. The administration of IQ-1S was associated with a faster amelioration of neurological status. Control rats showed a 2.0-fold increase in phospho-c-Jun levels in the hippocampus compared to the corresponding values in sham-operated rats 4 h after FCI. Administration of IQ-1S at a dose of 25 mg/kg reduced JNK-dependent phosphorylation of c-Jun by 20%. Our findings suggest that IQ-1S inhibits JNK enzymatic activity in the hippocampus and protects against stroke injury when administered in the therapeutic and prophylactic regimen in the rat model of FCI.

## 1. Introduction

C-Jun N-terminal kinase (JNK) is a critical mitogen-activated protein kinase (MAPK) activated by various brain insults and involved in neuronal injury triggered by reperfusion-induced oxidative stress [1]. Three distinct JNKs, designated as JNK1, JNK2, and JNK3, have been identified, and at least 10 different splicing isoforms exist in mammalian cells [2]. JNK3 is found almost exclusively in the brain [3], but it is not dominant, as JNK3 knockout results only in a weak attenuation of the total JNK pool in the brain tissue [4]. JNK phosphorylation and activity increase in the cerebral cortex and hippocampus after cerebral ischemia and reperfusion injury [5,6,7]. Sustained JNK activation has been shown to be associated with neuronal death and apoptosis following ischemic stroke and acute inhibition of JNK reduces infarction and improves outcomes in animal models of cerebral ischemia [7,8]. Because the inhibition of JNK isoforms has neuroprotective effects in animal models, it has been suggested that JNK inhibitors may represent promising therapeutic agents for the treatment of stroke.

A new specific JNK inhibitor, 11*H*-indeno [1,2-*b*]quinoxalin-11-one oxime sodium salt (IQ-1S), and several analogs of this compound have high affinities to JNK3 compared with their affinities to JNK1/JNK2 [9,10,11]. IQ-1S-treated mice demonstrate significantly reduced neurological deficit and infarct volumes as compared with vehicle-treated mice after 48 h of reperfusion [12]. The neuroprotective activity of IQ-1S may be due to the presence of NO-donating, antioxidant, hemorheological, endothelium-protective, and anti-inflammatory properties of the compound [12,13,14]. The ability of IQ-1S to inhibit JNK has been shown *in silico* and *in vitro* on cell cultures [9,10,11]. However, the JNK-inhibiting activity of IQ-1S has not yet been studied in the models of cerebral ischemia.

According to updated STAIR preclinical recommendations, the efficacy of the potential neuroprotector should be verified in at least two species using both histological and behavioral outcome assessments [15]. In this regard, the aim of our work was to demonstrate the neuroprotective activity of IQ-1S in the model of reversible focal cerebral ischemia (FCI) in rats and to reveal the potential JNK-inhibitory activity of the compound.

## 2. Materials and Methods

### 2.1. Animals

This study was carried out in accordance with the EU Directive 2010/63/EU concerning the protection of animals used for scientific purposes and was approved by the Animal Care and Use Committee of the Goldberg Research Institute of Pharmacology and Regenerative Medicine, Tomsk NRMC (protocol No. 130092017 from 08.09.17). Experiments were carried out on 52 adult male Wistar rats (weight: 250–280 g) obtained from the Department of Experimental Biological Models of the Goldberg Research Institute of Pharmacology and Regenerative Medicine. Rats were housed in groups of five animals per cage (57 × 36 × 20 cm) in standard laboratory conditions (ambient temperature of 22 ± 2 °C, relative humidity of 60%, and 12:12 h light–dark cycle) in cages with sawdust bedding, standard rodent feed (PK-120-1, Ltd., Laboratorsnab, Russia), and ad libitum water access.

### 2.2. Equipment

Rodent ventilator 7125 (Ugo Basile, Gemonio, Italy), temperature control unit HB 101/2 (Panlab Harvard Apparatus, Barcelona, Spain), homeothermic blanket control unit (Harvard Apparatus, Holliston, MA, USA), CO_2_ euthanasia device (Open Science, Moscow, Russia), and multipurpose centrifuge Scanspeed1248R (LaboGene, Allerød, Denmark) were used.

### 2.3. Chemicals and Drugs

Propofol-Lipuro (B. Braun Melsungen AG, Melsungen, Germany), diethyl ether for anesthesia (Kuzbassorgkhim, Kemerovo, Russia), Tween 80 (MERCK, Darmstadt, Germany), 10% neutral formalin (BioVitrum, St. Petersburg, Russia), 2,3,5-triphenyl tetrazolium chloride (Sigma-Aldrich, Saint Louis, MO, USA), and citicoline (Recognan, Alfa Wassermann S.p.A., Bologna, Italy) were used.

### 2.4. Study Molecule

The sodium salt of 11*H*-indeno[1,2-*b*]quinoxalin-11-one oxime (IQ-1S) (M314 series) was synthesized as described previously [16]. The chemical structure of IQ-1S was confirmed by the methods of massspectrometry and nuclear magnetic resonance; sample purity was 99.9%. To prepare IQ-1S emulsion, the weighted amount of IQ-1S powder corresponding to a proper dose for the animal was aseptically pounded with a pestle with 20 μL of Tween 80; 2.0 mL of physiologic sodium chloride solution was added to create a suspension.

### 2.5. Cerebral Ischemia Model

The FCI model was reproduced in rats by intraluminal occlusion of the left middle cerebral artery [17]. To manage general anesthesia with propofol (10 mg/kg/h), animals were implanted with a catheter placed in the right femoral vein under brief anesthesia with ethyl ether. FCI was achieved by using filaments (item #404145PK10Re) manufactured by Doccol Corporation (Sharon, MA, USA). The occlusion lasted for one hour; then, the filament was extracted; the external carotid artery was ligated, and the blood supply was resumed through the left internal carotid artery. The group of sham-operated animals received a similar surgical procedure, but without filament introduction. The body temperature of rats was maintained in a range of 37.0 ± 0.05 °C during the surgery by a temperature control unit and homeothermic blanket control unit. After wound closure and anesthesia recovery, animals were returned to their cages with free access to food and water.

### 2.6. Experimental Protocol

Two series of experiments were performed using a rat model of FCI. In series I, neurological status and infarct size were studied. Neurological status was assessed 4 and 48 h after modeling FCI. The size of cerebral infarction was assessed 48 h after reperfusion. Animals with FCI were assigned to five groups. The group of sham-operated animals (group 1) comprised five rats. Rats of the control group (group 2, *n* = 10) received 2 mL of physiological saline solution containing 20 µL of Tween 80 intraperitoneally 30 min prior to surgery as well as 24 and 48 h after the FCI procedure. Rats of the experimental groups (group 3, *n* = 10, and group 4, *n* = 9) received three intraperitoneal injections of 2 mL IQ-1S at doses of 5 and 25 mg/kg, respectively, in suspension, 30 min prior to surgery as well as 24 and 48 h after the procedure. The prophylactic/therapeutic scheme of IQ-1S administration was used, enablingdiscovery ofthe neuroprotective activity of this compound in the mouse model of FCI [12]. The animals received finaldoses of IQ-1S 48 h after FCI (day 2), neurological testing was performed 1 h later, and the animals were euthanized after that. Rats of group 5 (*n* = 8) received citicoline intraperitoneally at a dose of 500 mg/kg 15 min after the start of reperfusion and then after 24 h.

In series II, the inhibition of JNK 4 h after modeling FCI was studied. In this series, animals with FCI were assigned to three groups: sham-operated animals (group 6, *n* = 3), the control group (group 7, *n* = 3), and the experimental group (group 8, *n* = 4). Rats of the control and experimental groups received 2 mL of physiological saline solution containing 20 µL of Tween 80 or 2 mL of IQ-1S at a dose of 25 mg/kg intraperitoneally 30 min prior to surgery, respectively.

The rats were euthanized by CO_2_ euthanasia device in series I, and by decapitation 4 h after reperfusion in series II.

### 2.7. Neurological Deficit Evaluation

The neurological status of animals was examined by the experimenter unaware of the group the animals were assigned to. A degree of neurological abnormalities in the FCI model was determined by the method of Zhang H. et al. [18] based on the tail suspension test, maintenance of posture test, circling test, and horizontal righting reflex. The neurological deficit was characterized by a total score from all tests.

### 2.8. Assessment of Cerebral Infarct Size

To measure infarct size after FCI, the brain was frozen at a temperature of −12 °C for 2.5 h, and then frontal brain slices of 1.3mm in thicknesswere prepared using a series of histological knives assembled as a unit. After being thawed at room temperature, the brain slices were incubated in 0.5% solution of 2,3,5-triphenyltetrazolium chloride at 37 °C in dark for 15 min, and then the brain slices were fixed in 10% buffered formalin for 15 min, placed on glass slides, and scanned at600 dpi resolution by HP Scanjet 3770 (Hewlett-Packard, Beijing, China) with HP Director software v. 43.1.6.000. Images were stored in *.tiff format and processed with Adobe Photoshop 6.0 software. The cerebral infarction area and a total area of brain slice were calculated for each slide. The cerebral infarction area was expressed as a percentage of the total area of the slices.

### 2.9. Western Blot Analysis

Rats were euthanized by decapitation 4 h after reperfusion when the maximum activation of the JNK signaling system in the cerebral tissue and an increase in the content of P-MMK4, P-JNK and P-c-Jun may be observed [7]. The hippocampal tissues were isolated from the left hemisphere [19] on ice plate, homogenized and quickly frozen in liquid nitrogen. The tissue homogenate weighting 20 mg was ground with ice-cold RIPA lysis buffer containing 200 μL and phosphatase inhibitor cocktails and sonicated. Then the homogenate was centrifuged at 12,000× *g* rpm at 4 °C for 20 min, the supernatant was collected to determine the total protein concentration using the Quick Start Bradford Protein Assay (Bio-Rad, Hercules, CA, USA). Protein samples (50 μg) were diluted by SDS-loading buffer, boiled at 95 °C for 5 min, separated on 10% SDS-PAGE, and transferred to PVDF membranes (Immune-Blot, Bio-Rad, Hercules, CA, USA) using a semi-dry electrophoretic transfer system (Trans-Blot SD, Bio-Rad, Hercules, CA, USA). The membranes were then blocked for 1 h at room temperature in 5% BSA (Sigma-Aldrich, Saint Louis, MO, USA) in Tris-buffered saline (20 mM Tris and 150 mM NaCl) containing 0.1% Tween 20 and incubated with rabbit monoclonal anti-c-Jun (60A8) and anti-phospho-c-Jun (Ser63) (54B3) primary antibodies (both Cell Signaling, Danvers, MA, USA) diluted in the blocking solution (1:1000) overnight at 4 °C. They were then incubated with secondary horseradish peroxidase-coupled goat anti-rabbit IgG antibodies (Cedarlane, Burlington, ON, Canada) 1:10,000 in Tris-buffered saline solution containing 0.1% Tween 20 and 1% BSA for 1 h. Anti-β-actin-peroxidase monoclonal antibodies (Sigma-Aldrich, Saint Louis, MO, USA) were used as a loading control. Proteins were detected by chemiluminescent peroxidase substrate-1 (Sigma-Aldrich, Saint Louis, MO, USA) and the optical density of each band was determined by G:BOXChemiXRQ and Gene-Tools software version 4.3.8.0 (Syngene, Cambridge, UK). Values were normalized with respect to β-actin and expressed as a relative optical density.

### 2.10. Statistical Analysis

Statistical processing was performed with Statistica 8.0 software. All results are expressed as the mean ± SEM. Distributions of the quantitative variables were evaluated for normality by the Shapiro–Wilk W test. Group variation was assessed by the Kruskal–Wallis test and ANOVA. The significance of differences in the variables was evaluated by the Mann–Whitney Utest and ttest. Values were considered statistically significant when *p* was < 0.05.

## 3. Results

### 3.1. Effects of IQ-1S and Citicoline on Neurological Status in Rats with FCI

The surgical intervention did not lead to changes in the neurological status in group 1 of sham-operated animals. The development of neurological abnormalities in control group 2 was found immediately after anesthesia recovery. Muscle hypertonia of the forelimb occurred on a side contralateral to the lesion; animals were bending the body to the contralateral side after posture recovery. Four hours after FCI, the majority of animals demonstrated circlingonaspot, inability to extend the contralateral limb in a tail suspension test, and a significant decline in horizontal stability. A spontaneous decrease in the severity of the neurological deficit was observed following 48 h. For example, in the majority of animals, we observed the only inclination to the contralateral side during the movements; horizontal stability tended to recover after an initially significant decline; however, as a rule, an ability to extend the contralateral limb in a tail suspension test did not recover. The mean score of neurological deficit in the control group was 6 and 5, respectively, 4 and 48 h after the FCI procedure (Figure 1). In the control group, the mortality rate was 20%: one animal died on day 1 and another died on day 2.

All animals survived the entire experiment in group 3 with the administration of IQ-1S at a dose of 5 mg/kg. Total scores of neurological deficit decreased to 4 and 3 after 4 and 48 h, respectively, and were significantly lower than the control values (Figure 1). In group 4 administered with IQ-1S at a dose of 25 mg/kg, the mortality rate was 11%: one animal died on day 2. Significant decrease in a degree of neurological abnormalities compared to control value occurred in survived animals at 4 h and at day 2 after the onset of FCI; the mean score of neurological deficit decreased by 1.3- and 2.0-fold, respectively.

In group 5 with citicoline administration, one animal (12.5% of the total number of animals in the group) died on day 2. Total scores of neurological deficit in survived animals were 4 at 4 h, and 2 at 48 h after FCI (Figure 1). The neurological deficit scores in group 5 significantly differed at 4 and 48 h of observation compared with the corresponding index in the rats of the control group.

### 3.2. Effects of IQ-1S on Infarct Size in Rats with FCI

The results of studying the effects of IQ-1S and citicoline on the size of infarction in rats with FCI are shown in Figure 2a,b. Macrofocal cerebral infarction formed in all animals of control group 2 48 h after FCI. The infarct size was 25 ± 2% of the total area of brain sections. In rats of experimental group 3, infarct size was 20 ± 1% of the total area of brain sections, which was significantly lower than the corresponding parameter in the control group. The infarct size in experimental group 4 was 12 ± 1% of the total area of brain sections, which also significantly differed from the corresponding parameter in the control group. An assessment of infarct size in citicoline-treated rats of group 5 after FCI showed that the infarct size was 16 ± 4% of the total area of brain sections, which was significantly lower than the corresponding value in control group 2.

### 3.3. Inhibition of JNK

The results of Western blotting in series II are demonstrated in Figure 3a,b. A drastic 2.0-fold increase in the expression of phospho-c-Jun (*p* = 0.049) was observed in control rats (group 7) 4 h after FCI compared with sham-operated animals (group 6). Phospho-c-Jun in rats of experimental group 8 was lower than that in rats of the control group by 20% (*p* = 0.034). The expression of c-Jun did not significantly differ between groups 6, 7, and 8.

## 4. Discussion

In our study, intraluminal MCA occlusion in rats was used to model focal cerebral ischemia taking place in thrombotic stroke [20]. In this model, a zone of infarction forms and can be visualized. This model was used under the STAIR requirements stating that it is important to demonstrate that neuroprotective therapies reduce the infarct size [15]. While studying promising neuroprotectors, it is essential to assess the dose responses [15]. To study dose-dependent neuroprotective effects of IQ-1S, two doses of this compound were chosen: 25 mg/kg, which demonstrated a good neuroprotective effect in the experiments on mice with the same route of administration [12]; and 5 mg/kg, which was lower.

The dose-dependent effect of IQ-1S was found in both groups of rats administered with 5 and 25 mg/kg. Dose-dependent neuroprotective effects of IQ-1S wereobserved in the degree of infarct size reduction in the brain. Quantitative differences in the effects of IQ-1S at doses of 5 and 25 mg/kg were probably caused by the differences in the concentrations of the compound in the brain. Our previous pharmacokinetic studies [9] demonstrated the linearity of the IQ-1S pharmacokinetics.

Citicoline is used as a comparator agent for research and development of potential neuroprotectors [21,22]. The efficacy of citicoline in acute ischemic stroke is based on the stabilization of cell membranes, attenuation of glutamate excitotoxicity and oxidative stress, apoptosis inhibition, and elimination of endothelial dysfunction [23]. In our experiments, citicoline demonstrated a pronounced neuroprotective effect. Administration of citicoline and IQ-1S led to a decrease in necrosis zone on the periphery of the ischemic core and decreased the number of brain sections with necrosis. The effects of citicoline and IQ-1S did not significantly differ. A decrease in the infarction area in the brain of rats was accompanied by faster neurological status recovery. Due to the effects of citicoline, the neurological deficit was lower than control values by 1.5- and 2.0-fold on days 1 and 2 after FCI modeling, respectively. The extent of neurological deficit attenuation in both groups, administered with IQ-1S, corresponded to the effects of citicoline.

Our results confirm the efficacy of IQ-1S as a neuroprotective agent in the FCI model in rats. Similar results were demonstrated earlier for the FCI model in mice and for the global cerebral ischemia model in rats [12,13].

IQ-1S has a multitarget activity and a promising potential for neuroprotection. 

IQ-1S is a donor of exogenous nitric oxide (NO), which is produced through the conversion of the oxime group. An increase in NO concentration in the blood after intraperitoneal administration of IQ-1S [12] may contribute to the neuroprotective effect of the compound in the mouse model. NO, produced by the endothelial nitric oxide synthase (eNOS), may have neuroprotective properties [24]. In ischemia and reperfusion, NO production and the JNK signaling pathway are closely interrelated [25]. Donors of exogenous NO attenuate the S-nitrosylation of mixed-lineage protein kinase 3 (MLK3) and inhibit the activation of the JNK-dependent pathway after reperfusion [26]. NO donor sodium nitroprusside decreases JNK3 phosphorylation and the damage to hippocampal neurons after global ischemia-reperfusion [27]. The mechanisms of IQ-1S-mediated neuroprotection may involve the improvement of cerebral microcirculation caused by the better vasorelaxation, beneficial effects on blood viscosity, and attenuation of endothelial dysfunction. 

IQ-1S exerts antioxidant/antiradical IQ-1S activity [13]. Administration of IQ-1S significantly limits the lipid peroxidation products accumulation in brain tissue, which may also contribute to the neuroprotective effect after global cerebral ischemia. Antioxidant properties of IQ-1S may be mediated, in part, by the antiradical activity of the compound. A therapeutic approach using neuroprotective antioxidants is well justified by the concepts of cerebrovascular accidentpathogenesis [28], although clinical trials of individual drugs with free radical scavenging activity have not yet yielded results [29,30].

IQ-1S exerts anti-inflammatory activity. Recently, we found that macrophages and T cells could be immediate cell targets for IQ-1S-based anti-inflammatory immunotherapy [14]. After ischemic stroke, the integrity of the blood–brain barrier is compromised. Immune cells including T cells, B cells, dendritic cells, neutrophils, and macrophages infiltrate into the ischemic brain tissue and play an important role in regulating the progression of ischemic brain injury [31,32]. Thus, the immunomodulatory effects of IQ-1S may be involved in the mechanisms of neuroprotective properties of this JNK inhibitor.

IQ-1S alleviates high viscosity syndrome and attenuates endothelial dysfunction severity [13], whereas abnormal blood rheology and endothelial damage are essential for the pathogenesis of postischemic hypoperfusion [33,34].

All of the above mechanisms can significantly contribute to the integral neuroprotective effect. Considering that IQ-1S is a JNK inhibitor, JNK signaling downregulation during cerebral ischemia can be considered the main pharmacological effect of the compound, whereas its other properties may be pleiotropic. However, previous studies have not produced direct evidence for JNK inhibitory effects of IQ-1S in vivo.

JNKs are involved in many neuropathological signaling events and play a key role in the regulation of survival of the brain tissues in health and disease [35]. The signaling pathway of JNK plays a critical role in the mediation of apoptosis in cerebral ischemia and reperfusion [35]. Multiple works showed that increases in JNK phosphorylation and JNK-dependent signaling pathway activity are observed in the brains of rats and mice after the global and focal ischemia [4,36,37,38,39,40,41,42]. 

Various nonprotein synthetic inhibitors of JNK enzymatic activity are described including SP600125, AS601245, IQ-1S, SR-3306, and SU3327 as well as protein and nonprotein molecules inhibiting interactions of JNKs with their substrates, folding proteins, and/or cell organelles [9,35,38,43,44,45,46,47]. Some of them demonstrated neuroprotective activity in the animal models of stroke [7,48,49,50]. IQ-1S is one of the most specific small-molecule nonpeptide JNK inhibitors, which does not inhibit other kinases and IQ-1S has a high affinity to JNK3 compared with its affinity to JNK1/JNK2 [9,10]. 

The present work, for the first time, demonstrated the inhibitory effect of IQ-1S on JNK in brain tissue, which was confirmed by attenuation of the phospho-c-Jun increase in the rat hippocampus after FCI. Perhaps the ability of IQ-1S to selectively inhibit JNKs, especially JNK3, is one of the significant mechanisms of neuroprotective activity of the compound.

Obviously, the occurrence of neuroprotective effect requires the substance to efficiently permeate through the blood–brain barrier and to reach the concentration necessary for the effect manifestation in brain tissue. One of the disadvantages underlying a poor efficacy of the potential neuroprotectors consists of their low permeability through the blood–brain barrier [51]. Earlier, we [12] calculated the parameters of the IQ-1S molecule defining its ability to penetrate the blood–brain barrier: octanol/water distribution coefficient, the polar surface area of the molecule, and the number of rotatable bonds. Additionally, we performed a pilot pharmacokinetic study characterizing the intraperitoneal administration of IQ-1S at a dose of 25 mg/kg [52]. The results of the pharmacokinetic study allowed confirming previous theoretical calculations. After intraperitoneal administration of IQ-1S, the concentration of its active metabolite IQ-1 in the brain tissue started to increase from 30 min, reached 120 ng/g of tissue at 2 h and, then decreased and stabilized at the level exceeding 25 ng/g of tissue up to 8 h. The calculation shows that IQ-1 concentrations in brain tissue significantly exceed the concentration, which can inhibit JNK3 (K_d_< 10 nM) [10]. Therefore, one may suggest that the concentrations occurring in brain tissue after intraperitoneal administration of 25 mg/kg IQ-1S to rats are sufficient for JNK inhibition. 

## 5. Conclusions

IQ-1S exerted neuroprotective properties in the FCI model when introduced intraperitoneally in a therapeutic and prophylactic regimen to rats. The neuroprotective effect was present at doses of 5 and 25 mg/kg and was dosedependent. This effect consisted ofa significant reduction inthe infarction area in the brain. In terms of the effects on the neurological deficit and infarct size in the rat model of FCI, IQ-1S was noninferior to citicoline used as a positive control. The inhibitory effect of IQ-1S in the brain tissue was demonstrated as attenuation of the phospho-c-Jun increase in rat hippocampus after focal ischemia.

## Figures and Tables

**Figure 1 cells-09-01860-f001:**
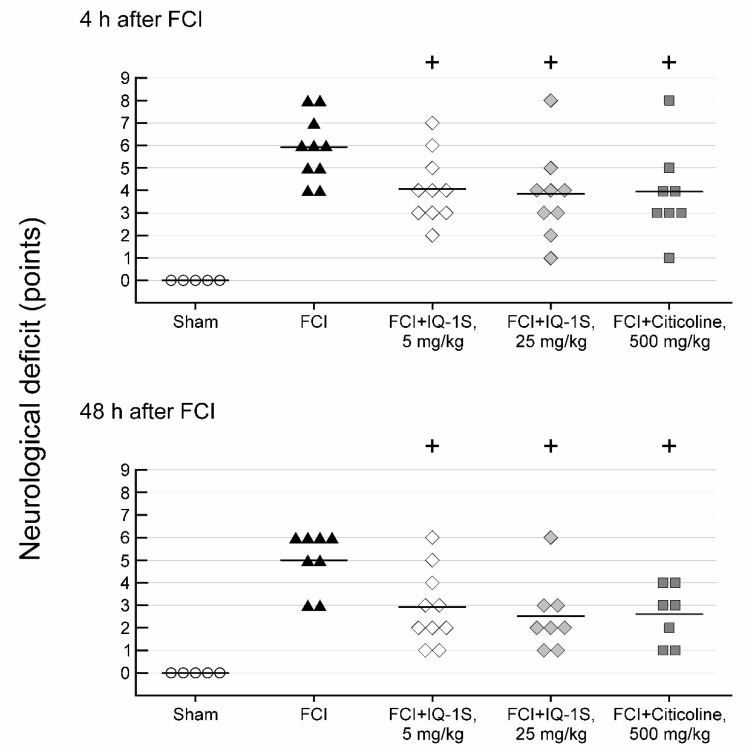
Effects of IQ-1S (30 min before, and 24 and 48 h after focal cerebral ischemia) and citicoline (15 min after the start of reperfusion and 24 h after focal cerebral ischemia) on neurological deficits. + *p* < 0.05, as compared with control animals.

**Figure 2 cells-09-01860-f002:**
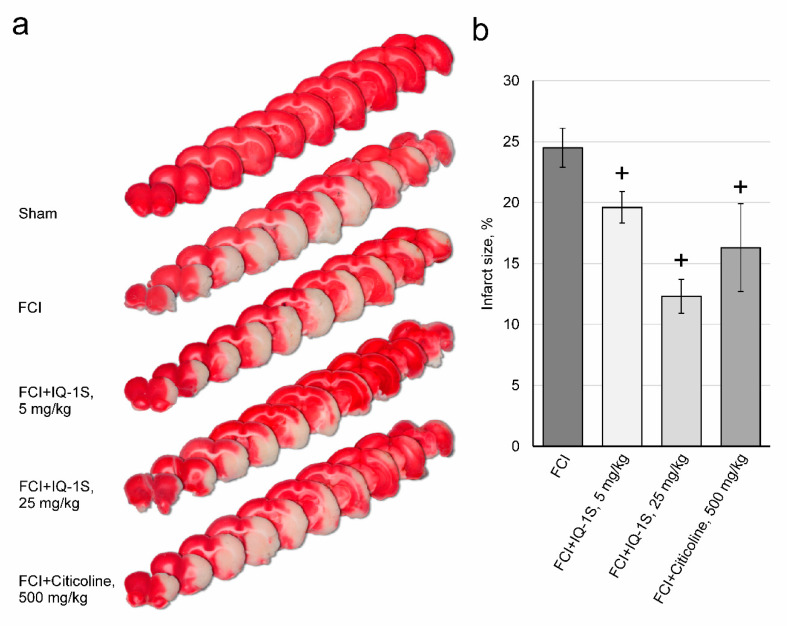
Effects of IQ-1S (30 min before, and 24 and 48 h after focal cerebral ischemia) and citicoline (15 min after the start of reperfusion and 24 h after focal cerebral ischemia) on infarct size in rat brains (**a**,**b**). White areas represent the infarct regions in the sections of representative samples of TTC-stained brain sections from rats sacrificed 49 h after FCI (**a**). + *p*< 0.05, as compared with control animals.

**Figure 3 cells-09-01860-f003:**
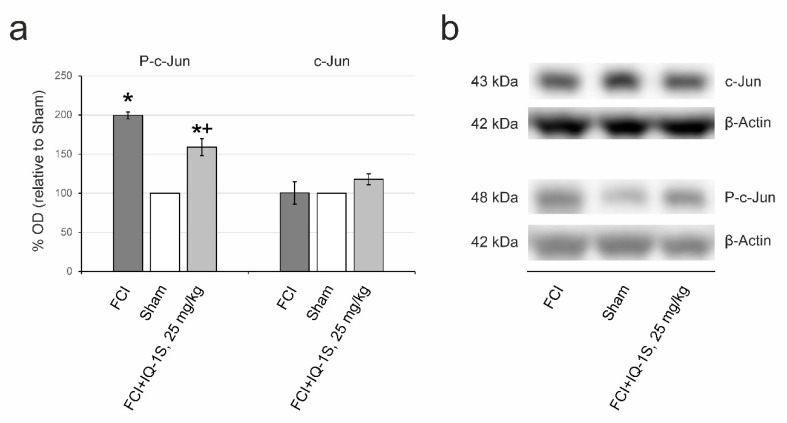
Effects of IQ-1S on total and phosphorylated c-Jun in the hippocampus 4 h after focal cerebral ischemia and reperfusion analyzed by Western blot. (**a**) Semiquantitative results of relative abundance of immunoreactivity for phospho-c-Jun (*p*-c-Jun) and c-Jun, as determined using densitometric measurement on immunoblots. (**b**) Representative immunoblot bands in each group. * *p* < 0.05, as compared with sham-operated animals; + *p* < 0.05, as compared with control animals.

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
