# Peer review of "Neuroprotective Effects of a Novel Inhibitor of c-Jun N-Terminal Kinase in the Rat Model of Transient Focal Cerebral Ischemia"

_cells, 2020, doi:10.3390/cells9081860_

Round 1

Reviewer 1 Report

Activation of the C-Jun N-terminal kinase (JNK1-3) pathway during cerebral ischemia-reperfusion has been linked to neuronal injury. Hence, brain-permeable JNK inhibitors may provide a therapeutic approach for the treatment of stroke.  The manuscript by Plotnikov et al continues the series of publications by the team on neuroprotective effects of a novel selective JNK3 inhibitor IQ-1S in several rodent models of cerebral ischemia. Using the rat model of transient focal cerebral ischemia, the authors now demonstrate that IQ-1S (5 and 25 mg/kg ip) administered in combinatorial therapeutic and prophylactic regimens, reduced the infarct size and improved the neurological recovery 24 and 48 h after stroke.  Brain JNK-targeting effects of IQ-1S were confirmed by reduced c-Jun phosphorylation in the post-stroke hippocampus. The authors conclude that IQ-1S inhibits JNK enzymatic activity and protects against stroke injury.  Overall, this well-written manuscript adds to previously accumulated information on therapeutic potency of IQ-1S in ischemic stroke treatment.  However, I have several concerns regarding the experimental planning and the quality of supporting data.

Major concerns.

  1. The IQ-1S treatment protocol needs additional substantiation. It is not clear why the IQ-1S treatment included a combination of prophylactic (30 min prior to stroke) and therapeutic (24 and 48 h) dosages of the drug. The neuroprotective effects of IQ-1S in stroke have been demonstrated by the team in their previous publications. Detecting the therapeutic window of post-stroke IQ-1S would be of outmost importance for clinical implications of the drug.
  2. The neurological deficit methodology (overall score) needs to be described in detail.
  3. Figure 2: High resolution images of post-stroke histology sections are not shown.
  4. Low quality of Western Blot images for P-cJun/cJun is unacceptable (Fig. 3b).
  5. Considering that IQ-1S is a selective JNK3 inhibitor, detecting JNK3 versus JNK1/JNK2 activities in the experimental groups is necessary.
  6. The detection of JNK1-3 activities in different post-stroke brain areas will provide important information regarding the mechanism of neuroprotection.  
  7. The authors previously demonstrated that IQ-1S is a nitric oxide (NO) donor (12). The contribution of NO to neuroprotective effects of IQ-1S in the rat model of stroke needs to be addressed.

Minor concerns.

  1. Figure 1, X-axis: “Neurological deficit” seems to be more appropriate than “Neurological score”.
  2. Figures 1 and 2 legends. Please indicate the IQ-1S treatment protocol: one dose before and two doses after stroke (24 and 48 h)?
  3. It is not clear whether the animals were sacrificed immediately after a 48-h dose of IQ-1S for detection of neurological deficits and the infarct size (Figs. 1 and 2)? If so, what was the purpose of the 48-h treatment?

Reviewer 2 Report

In this manuscript ‘Neuroprotective effects of a novel inhibitor of c-Jun N-terminal kinase in the rat model of transient focal  cerebral ischemia’ the authors were investigated the mechanisms of neuroprotective activity of IQ-1S in the models of reversible focal 25 cerebral ischemia (FCI) in Wistar rats.

Since there has not been a lot of options for very effective treatment after cerebral ischemia, every attempt to find a new cure is beneficial.

The paper is well written in a concise manner.  Nevertheless, I would suggest the following recommendations.

Introduction: authors could add a relevant information about JNK phosphorylation and JNK activity in cerebral cortex since major cases of cerebral ischemia affects this structure of the brain.

Method. Author could comment the choise of diethyl ether for anesthesia since other options such as isoflurane has less effect on animal physiology. Add number (size) of filament used for occlusion of the left middle cerebral artery. Indicate and was given to animal for the surgery for the pain relief. Briefly describe post-operative care of animals. Short description of behavioural test (the tail suspension test, posture maintenance, circling test, and horizontal righting reflex) are essential in this section. It is not clear how neurological deficit is calculated.

Results. Data of all behavioral test (the tail suspension test, posture maintenance, circling test, and horizontal righting reflex) should be presented and then can be visualized in summary picture as a neurological deficit.

Authors state that ‘No significant differences with the control group were observed 4 h after FCI: the mean score of the neurological deficit was 4’ could be that all animals are still under effect of anesthesia?

The figure representing WB blots Figure 3B is poor quality. This has to be submitted in better quality.

Discussion: Author should discuss why inhibition of JNK using WB was chosen to perform (only) 4h after cerebral ischemia but not 48h (the neuroprotective effect of IQ-1S was pronounced. Why authors did not decided to test inhibition of JNK using WB also for cerebral cortex since the infarct size mainly affects this structure?

Reviewer 3 Report

This study by M.B. Plotnikov et al., explored neuroprotective potential of the JNK3 inhibitor IQ-1S in a rat model of the middle cerebral artery occlusion (MCAO).  Similar to several other tested in the literature JNK inhibitors, IQ-1S reduced ischemic infarction and alleviated neurological deficits.  

Overall, this work appears to be solid albeit not highly innovative.  The neuroprotective properties of IQ-1S have been already demonstrated in rodent models of stroke, including MCAO in mice (Atochin et al., Neurosci Lett 2016) and global ischemia in rats (Plotnikov et al., Molecules, 2019).  On its own, the redundancy of this work is not a big issue since the high reproducibility of neuroprotection findings and consistency of effects in different stroke models are sought after in the cerebral ischemia field (this is important but rather small part of STAIR criteria).  However, I have some concerns about presentation, analysis, and interpretation of the newly collected information.  The Authors need to address these issues to bring their work in line with publication standards.

Major points:

[1] One major concern is whether the Authors accurately describe and interpret the mechanism of action of the IQ-1S compound.  In their previous work, they found that systemic delivery of IQ-1S dramatically elevated blood nitrite levels (Atochin et al., Neurosci Lett 2016) and reduced blood viscosity/coagulation (Plotnikov et al., Molecules, 2019).  Furthermore, IQ-1S demonstrated intrinsic antioxidant properties (Plotnikov et al., Molecules, 2019).  On their own, these off-target effects are enough to reduce MCAO infarction volume due to enhanced collateral circulation, prevention of secondary clotting, and alleviation of oxidative damage within the ischemic territory.  Unless the Authors have much better, JNK3-related controls, they need to disclose the alternative mechanisms of IQ-1S actions in the Introduction and START Discussion with consideration of plural mechanisms of neuroprotection.

[2] The Authors need to double-check their statistical analyses.  As a test, I have re-run ANOVA and Kruskal-Wallis ANOVA on the results presented in Fig. 1 for the 4-hours group and found no statistical differences for any of the treatment groups, but particularly for 25 mg/kg IQ-1S.  Based on this, I have a concern about accuracy of the analytical approach in this paper.  In order for this manuscript to be accepted, I recommend requesting submission of all primary data included in Figures 2, 3 and their reevaluation of the journal-appointed statistician.

[3] Representative western blot images in Fig. 3B are highly pixilated and appear to be assembled from separate “mini-panels”.  In order to avoid any undesirable questions, I recommend to the Authors to provide higher resolution western blots and include full size images for c-Jun and p-c-Jun immunoreactivity as a supplement.  Frankly, I think that for the declared three protein samples per group presented error bars in WB analysis are unrealistically small.  Also, selection of hippocampal tissue for this assay is very odd because hippocampus is not supplied by MCAO and cannot be reliably considered a good representation of penumbral tissue.  Overall, this is not a small issue because the Authors use this bit of information as “innovative”. 

Minor-to-moderate concerns:

[4] The English in this manuscript requires additional editing.

[5] Reference 23 to unpublished work is unacceptable.

[6] Use of citicoline as a “comparator” in neuroprotection studies is Ok but not really a good benchmark since citicoline failed in clinical trials.

Round 2

Reviewer 1 Report

The key conclusion of the manuscript that IQ-1S provides neuroprotection by inhibiting JNK activation in stroke remains not fully substantiated.

The comment on unacceptable quality of Western Blot images for P-cJun/cJun (Fig. 3) has not been addressed appropriately.  There is a discrepancy between a representative WB (Fig. 3b), quantitative data (Fig. 3a), and the data interpretation (P. 6). It is not clear how many animals were used for P-cJun/cJun detection in the hippocampus. Total c-Jun has been greatly reduced in the FCI group (Fig. 3b),  although the authors state that "The expression of c-Jun did not significantly differ between groups 6, 7, and 8” (P. 6). A good quality WB is required to convince the reader that there is a 20% reduction (P<0.05) in P-c-Jun in IQ-1S and to support the key conclusion that IQ-1S provides neuroprotection by inhibiting JNK activation.  Other mechanisms, such as NO contribution to neuroprotective effects of IQ-1S, might be of greater importance.

Author Response

Dear Reviewer I,
Thank you for your valuable comments and suggestions contributing to the better quality of our manuscript. Please find below our responses to the comments.
A discrepancy between the representative quantitative Western Blot (WB) data (Fig. 3a: c-Jun, FCI) and images (Fig. 3b: c-Jun, FCI) is corrected. In the new version of Figure 3b, WB images from rat #3 (control group) are replaced by the images from rat #2 from the same group.
The number of animals used to study the effect of IQ-1S on P-cJun/cJun in the hippocampus is indicated in section 1.6 Experimental protocol (series II).
We agree that other mechanisms, such as NO contribution to the neuroprotective effects of IQ-1S, might be of greater importance. The relevant information was added to the Discussion.

Reviewer 2 Report

Thank you for the answers provided.

I still recommend to consider for the next stroke studies to use post-operative care that include hydrogel and powder form of (made wet with water) diet for stroke animals it will increase animal survival and lab will follow 3Rs.

Author Response

Dear Reviewer 2,

Thank you for your recommendation for post-operative care that we will consider for our future stroke studies.

We thank all reviewers for important comments about the quality of English. We did our best to correct the manuscript for English.

Reviewer 3 Report

I am satisfied with the Authors' responses.

Author Response

Dear Reviewer 3,
Thank you for your valuable comments and suggestions contributing to the better quality of our manuscript.